# MRI Radiomics and Predictive Models in Assessing Ischemic Stroke Outcome—A Systematic Review

**DOI:** 10.3390/diagnostics13050857

**Published:** 2023-02-23

**Authors:** Hanna Maria Dragoș, Adina Stan, Roxana Pintican, Diana Feier, Andrei Lebovici, Paul-Ștefan Panaitescu, Constantin Dina, Stefan Strilciuc, Dafin F. Muresanu

**Affiliations:** 1Department of Neurosciences, Iuliu Hațieganu University of Medicine and Pharmacy, No. 8 Victor Babeș Street, 400012 Cluj-Napoca, Romania; 2RoNeuro Institute for Neurological Research and Diagnostic, No. 37 Mircea Eliade Street, 400364 Cluj-Napoca, Romania; 3Neurology Department, Emergency County Hospital, No. 43 Victor Babes Street, 400347 Cluj-Napoca, Romania; 4Department of Radiology, Iuliu Haţieganu University of Medicine and Pharmacy, No. 3–5, Clinicilor Street, 400006 Cluj-Napoca, Romania; 5Department of Microbiology, Iuliu Hatieganu University of Medicine and Pharmacy, No. 8 Victor Babes Street, 400012 Cluj-Napoca, Romania; 6Department of Radiology, Faculty of Medicine, Ovidius University, 900527 Constanta, Romania

**Keywords:** radiomics, ischemic stroke, predictive model

## Abstract

Stroke is a leading cause of disability and mortality, resulting in substantial socio-economic burden for healthcare systems. With advances in artificial intelligence, visual image information can be processed into numerous quantitative features in an objective, repeatable and high-throughput fashion, in a process known as radiomics analysis (RA). Recently, investigators have attempted to apply RA to stroke neuroimaging in the hope of promoting personalized precision medicine. This review aimed to evaluate the role of RA as an adjuvant tool in the prognosis of disability after stroke. We conducted a systematic review following the PRISMA guidelines, searching PubMed and Embase using the keywords: ‘magnetic resonance imaging (MRI)’, ‘radiomics’, and ‘stroke’. The PROBAST tool was used to assess the risk of bias. Radiomics quality score (RQS) was also applied to evaluate the methodological quality of radiomics studies. Of the 150 abstracts returned by electronic literature research, 6 studies fulfilled the inclusion criteria. Five studies evaluated predictive value for different predictive models (PMs). In all studies, the combined PMs consisting of clinical and radiomics features have achieved the best predictive performance compared to PMs based only on clinical or radiomics features, the results varying from an area under the ROC curve (AUC) of 0.80 (95% CI, 0.75–0.86) to an AUC of 0.92 (95% CI, 0.87–0.97). The median RQS of the included studies was 15, reflecting a moderate methodological quality. Assessing the risk of bias using PROBAST, potential high risk of bias in participants selection was identified. Our findings suggest that combined models integrating both clinical and advanced imaging variables seem to better predict the patients’ disability outcome group (favorable outcome: modified Rankin scale (mRS) ≤ 2 and unfavorable outcome: mRS > 2) at three and six months after stroke. Although radiomics studies’ findings are significant in research field, these results should be validated in multiple clinical settings in order to help clinicians to provide individual patients with optimal tailor-made treatment.

## 1. Introduction

Stroke is a leading cause of mortality and disability, resulting in substantial socio-economic costs for post-stroke care [1,2]. Although the mortality rates have declined over the past two decades, the absolute number of incident stroke, disability-adjusted life-years lost due to stroke, and stroke-related deaths is increasing [2].

Predictive models (PMs), which integrate patient characteristics and care process to estimate the probability of developing a particular event or future outcome have been proven valuable in the primary prevention of cerebrovascular diseases [3]. PMs such as Framingham Score [4], QRISK [5], Reynolds [6] and Euro-Score [7] have been used in cardiovascular and cerebrovascular diseases to help health service planning and to support clinical decision making, diagnostic and therapeutic management in risk groups. A systematic review [3] of 109 studies on clinical PMs for functional outcome in ischemic stroke concluded that, in the thirty-five years of literature, the following clinical factors are consistently identified as the most suitable predictor variables of functional outcome and mortality: age, gender, stroke severity, stroke subtypes and comorbidities such as diabetes and atrial fibrillation. Ntaios et al. [8] demonstrated that recently introduced prognostic scores such as ASTRAL [9], DRAGON [10] and SEDAN [11] predict outcome of AIS patients with higher accuracy compared to clinical predictions made by physicians, providing evidence that PMs may positively impact patient outcome. All three scores [8] incorporate age, admission National Institute of Health Stroke Scale (NIHSS) and blood glucose level as predicting variables, whereas DRAGON [10] and SEDAN [11] contain as predictive feature hyperdense middle cerebral artery (MCA) sign or early infarct signs on computer tomography (CT). During the past decade, advances in computational technologies, especially in machine learning, have placed medical imaging in an increasingly central role in patient-specific management [12]. This progress makes it possible to convert subjective visual interpretation into an objective assessment that is driven by image data [12]. 

Radiomics analysis (RA) has emerged in this context, being a method that extracts undiscovered imaging features by converting routinely acquired images into higher dimensional data [13,14,15,16]. This process is motivated by the concept that digitally encrypted images contain information related to the pathophysiology of certain diseases, and this information can be exploited via quantitative image analysis [13]. Currently, in the ischemic stroke field, the role of RA was explored in three domains: diagnosis of stroke lesion, prediction of early outcome and long-term prognosis assessment [12]. The diagnostic role of radiomics in stroke lesions was investigated using CT or magnetic resonance imaging (MRI). Oliviera et al. [17] performed texture analysis (TA) on non-contrast CT images of acute ischemic stroke (AIS) patients to distinguish healthy tissue from regions affected by AIS and found that TA parameters were significantly different between patients and controls, with the most discriminative feature being angular second moment. By using MRI, Sikio et al. [18] assessed 30 patients with chronic right hemisphere stroke and found that the ischemic region had lower homogeneity compared with non-affected side and relatively high values of complexity and randomness. Ortiz-Ramon et al. [19] used multimodal MRI data of different brain regions from 100 patients to investigate if RA could distinguish between patients who had prior ischemic stroke and the stroke-free health population. They showed that TA and wavelet transformation could identify the presence of previous stroke lesions with favorable discrimination (area under the ROC curve (AUC) > 0.7) independently on what MRI sequence has been used or what brain region has been affected [19]. Regarding early outcomes after AIS, Kassner et al. [20] investigated if RA could predict hemorrhagic transformation in AIS patients treated with intravenous thrombolysis and suggested that radiomics features could be a better predictor compared to visual enhancement score in post-contrast T1-weighted MRI (AUC > 0.75 compared to AUC < 0.6). Qiu et al. [21] conducted RA to predict early recanalization after proximal occlusion in large vessels in 67 AIS patients treated with intravenous thrombolysis and suggested that the combination of RA features from non-contrast CT and CT angiography was more predictive of early recanalization with an AUC of 0.85 compared with conventional thrombus imaging features such as length, volume or permeability. Regarding post-stroke cognitive impairment, Betrouni et al. [22] showed that texture features of hippocampus and entorhinal cortex at 72 h after AIS onset can predict the occurrence of cognitive impairment. Their results were further confirmed in a rat model of middle cerebral artery occlusion, with significant correlation being demonstrated between texture features and hippocampal neural density [22]. The increasing number of studies investigating RA and machine learning algorithms applications in ischemic stroke with variable protocols and design allows for data pooling. 

This review aims to systematically evaluate the role of RA in acute ischemic stroke neuroimaging and the potential applications in clinical practice. The primary objective is to compare the results of AIS studies using RA for clinical outcome prediction. The secondary objective is to assess the methodological quality of studies using radiomics quality score (RQS).

## 2. Materials and Methods

The PRISMA (Preferred Reporting Items for Systematic Reviews and Meta-analysis) [23] statement was used for this systematic review (Appendix A Appendix A). The protocol for this review is available in the OSF registry, https://osf.io/9dx6j/ accessed on 31 January 2023. Before a formal search was conducted, we used the keywords to perform preliminary search stage in several preprint and peer-reviewed databases. The selection of databases depends on the availability of data and the degree of overlap between databases. Publications in English assessing MRI radiomics features in AIS patients published from the earliest date available until our last search date of 31 December 2022 were searched on two electronic databases (PubMed and Embase). The search terms consisted of MRI, radiomics and stroke. The detailed search string is displayed in the Appendix A (Appendix A). Two researchers assessed the eligibility of the articles through title and abstract screening using the inclusion and exclusion criteria (Table 1). Any disagreements were resolved by consensus. The full text of articles in which RA was applied on MRI images of AIS patients for predictive purposes were obtained for further evaluation. 

Although CT seems to be the most commonly used technique for RA modeling [12], several studies [24,25,26] have recently suggested that the majorities of radiomics features are highly affected by image acquisition and reconstruction parameters and thus their reproducibility could be affected. Moreover, a phantom study [27] showed that diverse CT scanners made by different manufactures could cause variability in RA values. Thus, we selected the studies which performed RA on MRI images.

Lohman et al. [16] proposed a list of recommendations that should be considered in study investigating the value of radiomics in research or clinical practice, from preferred methods for quality evaluation to radiomics workflow steps that should be reported. Thus, we created a specific standardized data extraction form consisting of the following categories: image acquisition, image pre-processing, segmentation, feature extraction, feature selection, model generation and validation, model testing, results, and clinical translation. All of these categories are addressed within the radiomics quality score (RQS) [15,16], which is a tool developed to assess the methodological quality of studies using radiomics [15,28]. Thus, we chose to use RQS to analyze the main radiomics steps among studies and to present the extensive RA process for each study only in Appendix A–Appendix A.

The detailed RQS score is described in the Appendix A–Appendix A. Two readers with previous experience in radiomics independently assigned an RQS score to each article included in this systematic review. The reviewers extracted the data using a predefined RQS form used in other systematic reviews on RA [28,29,30] according to RQS six domains [15]: protocol quality and reproducibility, feature reduction and validation, clinical validation and utility, the performance index, high level of evidence, and open science with open availability of source code and data. Any disagreement was resolved by consensus. The total RQS score was calculated for each article and for each component (score range, −8 to 36) and expressed as a median and interquartile range. For the six domains in the RQS score, basic adherence was assigned. 

The main goal of radiomics is to establish a practical and accurate model for predicting clinical outcomes [12]. A prediction model is defined as any combination of 2 or more predictors (demographic, clinical, imaging or biological variables) for estimating for an individual the probability of developing a particular outcome [31]. Therefore, we extracted the studies’ data regarding the predictive models employed using the following categories: model objective, clinical features, conventional imaging features, biological features, radiomics features, validation methods, main results, and limits.

PROBAST [31,32] was designed for use in systematic review or prediction model studies and consists of four domains (participants, predictors, outcome and analysis) containing twenty signaling questions to facilitate risk of bias assessment. A graphical summary presenting the percentage of studies rated by level of concern (low risk of bias, high risk of bias, unclear risk of bias) was displayed. The data were reported in a qualitative narrative synthesis based on the identified categories. The results’ risk of bias and applicability were compared with existing literature. Unfortunately, the studies were methodologically heterogeneous, and meta-analysis was not possible.

## 3. Results

In total, 150 articles were obtained, out of which 36 were duplicates. Of the remaining 114, 87 were rejected during title and abstract screening. Twenty-one articles were eligible for full-text evaluation. Six articles fulfilled the pre-established eligibility criteria. The study selection process is displayed in the PRISMA flow diagram [23] (Figure 1), whereas Table 2 contains details on study design, characteristics of study population, clinical and imaging variables integrated in PMs and the performance of PMs for each study. 

All studies [33,34,35,36,37,38] investigated the predictive value of radiomics features in assessing AIS clinical outcome. Clinical outcome was evaluated at ninety days in five studies [33,34,36,37,38], respectively, at six months in one study [35] using the modified Rankin scale (mRS) and the patients were dichotomized into good outcome (mRS = 0, 1, or 2) and poor outcome (mRS = 3, 4, or 5) groups. Additionally, one study [37] assessed the role of radiomics-based models for predicting one-year ischemic stroke recurrence confirmed on diffusion-weighted imaging (DWI).

Five studies [33,34,35,36,37] integrated separately clinical and radiomics features and then combined these variables within PMs, tested its performance and validated into another datasets. Additionally, two studies [33,34] used conventional MRI features such as infarct volume, orthogonal diameters of ischemic lesion, DWI-Alberta Stroke Program Early CT Score (ASPECTS) or Fazekas score together with clinical and RA parameters. Among the clinical factors known to be independent predictors for AIS outcome, the most used in the PMs were age, gender, admission NIHSS, 24-h NIHSS, prior documented stroke, atrial fibrillation, hypertension or diabetes [33,34,35,36]. These studies also conducted preliminary univariate and multivariate analysis to select the clinical features that were significantly associated with unfavorable outcome. Additionally, an interclass-correlation coefficient with a cut-off of 0.75 was used to evaluate the consistency between the researchers for estimating infarct volume and admission NIHSS [34] and to assess the reliability of extracted RA features [33,34,35,36,37]. The most used MRI sequences for feature extraction were apparent diffusion coefficient (ADC) [33,35,36,38] and DWI [34,35,37]. In all studies, the region of interest was the ischemic lesion which underwent manually segmentation performed by at least two experienced neuroradiologists [33,34,35,36,38]. Only one study [37] applied automatic segmentation. The number of radiomics features extracted varied from 15 [38] to 1310 features [35], but after applying feature reduction methods, the number decreased at 6 [33] to 100 [37], respectively. Most of the studies [33,34,35,36,37,38] used first-order statistics and second-order statistics (texture analysis), but three studies [33,36,37] applied high-order statistics, such as wavelet or Laplacian of Gaussian transformation, respectively. Three studies [34,35,37] were from single center and built validation cohorts from the same institute, whereas only one study [33] applied the PM to datasets from two different institutes. The description of radiomics workflow for each study is depicted in Appendix A–Appendix A. 

Five studies [33,34,35,36,37] evaluated predictive performance for different PMs. Three studies [33,34,35] initially investigated models based only on clinical factors and the most performant PM [35] consisted of clinical variables such as age, gender, stroke history, diabetes, baseline mRS and NIHSS, achieving an AUC of 0.82 (95% CI, 0.77–0.87). Additionally, one study [33] built a PM based on clinical and conventional MRI features such as age, gender, admission NIHSS, DWI-ASPECT score and orthogonal diameters of infarct lesion and obtained an AUC of 0.78 (95% CI, 0.68–0.88). One study [33] compared radiomics-based PMs depending on the MRI sequence used for feature extraction and showed that ADC radiomics-based PM seems to achieve a better predictive performance compared to FLAIR radiomics-based PM (AUC = 0.77, 95% CI 0.62–0.83 versus AUC = 0.73. 0.62–0.83). Moreover, when ADC and FLAIR radiomics features were added in the same PM, the predictive value was higher (0.81, 95% CI 0.73–0.89) [33]. In all studies [33,34,35,36,37], the combined PMs consisting of clinical and imaging features have achieved the best predictive performance compared to PMs based only on clinical or only on radiomics features, with the results varying from an AUC of 0.80 (95% CI, 0.75–0.86) [34] to an AUC of 0.92 (95% CI, 0.87–0.97) [33]. The best PM [33] was validated in external datasets from two different institutes, obtaining an AUC of 0.864 (95% CI, 0.773–0.954) in the validation cohort.

Two studies [34,35] developed a radiomics- and clinical-based nomogram, which is an easy-to-use scoring model with the ability to assess the risk of unfavorable outcome in individual patients [39]. Wang et al. [34] included in their nomogram clinical variables such as age, 24-h NIHSS or the presence of hemorrhagic transformation and 11 radiomics features, reaching an AUC of 0.80 (95% CI 0.75–0.86) in the training cohort and an AUC of 0.73 (95% CI 0.64–0.82) in the validation set. On the other hand, Zhou et al. [35] created a nomogram with higher performance (AUC = 0.868, 95% CI 0.825–0.910 in the training set and AUC = 0.890, 95% CI 0.844–0.936 in the validation set), including the following features: age, gender, prior stroke, baseline NIHSS, baseline mRS, diabetes and 7 radiomics features. The previous study of Wang et al. [38] did not find a predictive value of texture features in assessing the stroke outcome but demonstrated that ADC-entropy and T2-FLAIR 0.75 quantile have predicted AIS severity with an AUC = 0.7 (*p* = 0.01).

Regarding the methodological quality of the six radiomics studies, the median RQS score was 15 (interquartile range, 4), which represented 36.11% of the ideal score of 36 [15]. The adherence rate of the RQS of all included studies is depicted in Figure 2. The RQS assessment for each study is described in Appendix A–Appendix A. The lowest score was 6 and the highest score was 16. The RQS of selected studies was lowest in the following domains: high level of evidence, open science, and model performance index (Figure 2), meaning that the most of studies did not validate the results in further prospective cohorts, did not perform a cost-effectiveness analysis of the model, did not make the code or radiomics data publicly available and did not use calibration and cut-off analysis in order to promote model reproducibility. Meanwhile, studies [33,34,35,37] with higher RQS earned additional points by using multiple segmentations or external validation based on datasets from distinct institutes.

Regarding the risk of bias assessment, the PROBAST [32] tool was used. The overall risk of bias based on the four domains of PROBAST depending on three levels of concern (low, high or unclear risk of bias) is depicted in Figure 3. The PROBAST assessment for each study is described in Appendix A–Appendix A. Both overall risk of bias and applicability of concerns were low. Two studies presented high risk of bias regarding participant selection, excluding a large number of patients from the initial cohorts due to comorbid diseases that may affect their long-term stroke outcome. Unclear risk of bias due to unavailable information regarding the predictors and outcome analysis was established in the case of one study.

## 4. Discussion

Prognostic scores may not fit to all cohorts due to patients’ differences regarding racial or ethnic identity, background or comorbidities, hospital type or healthcare system, and acute stroke management [40]. Poststroke functional outcome is affected by a variety of factors, such as age, gender, comorbid diseases, stroke severity, stroke subtypes, and treatments before and after discharge [41,42,43]. Age and stroke severity are considered significant factors [40], which is consistent across the majority of studies assessing predictive scores or PMs, even those based on automatic algorithms [33,34,35]. 

The external validity of initial stroke prognostic scores is limited [40,44]. A recent study [44] on 10,777 patients investigating eight stroke prognostic clinical scales confirmed differences in the prognostic accuracy when they are applied to external datasets, suggesting that even the best performing scale had a prognostic accuracy that may not be sufficient as a basis for clinical decision-making. 

In the era of large amounts of data and artificial intelligence (AI), automated systems may be helpful in predicting outcomes in patients with stroke and providing individual patients with optimal tailor-made treatment [40]. The current applications of AI in AIS field seem to be efficient in numerous parts of the diagnostic and management pathways, including detection, triage, and outcome prediction [45]. Computer-aided detection schemes based on texture features from areas known to show early AIS signs such as insula ribbon and lentiform nucleus were suggested to be a promising algorithm for lacunar AIS diagnosis [46]. As lacunar AIS is relatively difficult to diagnose on CT within the first hours after onset [47], early detection is crucial for establishing the best treatment, and there is a need for a more efficient method to improve CT detection rate. Automated color maps (e.g., ColorViz) have proved to be rapid and accurate post-processing tools that permit maintenance of the temporal resolution of CT angiography, summing in a single image the three different cerebral vascular phases using a time variant color map [48,49]. As the definition of the collateral circulation status is essential in selecting patients for mechanical thrombectomy, the possibility of using an immediate scoring scale for CT angiography could make the diagnostic assessment faster and easier. A recent systematic review [50] showed that AI-based comprehensive platforms (e.g., Brainomix, iSchemaView, Viz.ai) could automatically detect the presence of large vessel occlusion (LVO), being a catalyst for timely LVO detection and an aid to acute management decision-making. Moreover, automated clot composition analysis systems using machine learning seem to provide information on the cause of cerebral artery occlusion and may further guide acute revascularization and secondary prevention. For example, a recent study [51] assessed the accuracy of a such algorithm based on blooming effect on pre-treatment gradient echo images (GRE) from 67 patients with middle cerebral artery stroke and identified atrial fibrillation with high accuracy (AUC > 0.87). Blooming artifacts caused by paramagnetic materials in GRE images have been associated with cardioembolic stroke [52,53], cardioembolic clots having significant higher proportion of red blood cells compared with noncardiac clots and, oxyhemoglobin in erythrocytes goes through sequential stages of degradation into deoxyhemoglobin and hemosiderin, which are paramagnetic materials [54,55]. Conventional MRI parameters extracted from DWI and fluid-attenuated-inversion recovery (FLAIR) sequences had been proven to be significant predictor of stroke outcome [56,57,58,59]. Recent evidence suggest that DWI lesion may not be entirely composed of irreversibly damaged core. A systematic review [60] on tissue outcome of DWI hyperintense stroke lesions suggested that hyperintense DWI lesions are rather heterogenous regions comprising various biochemical and metabolic environments, which may be variably amenable to salvage rather than as homogenous regions of ischemic core tissue. Guadagno et al. [61] investigated oxygen metabolism in DWI lesions and revealed spatial variability in the cerebral metabolic rate of oxygen with individual DWI lesions. Additionally, significant variability of oxygen extraction fraction was demonstrated within single DWI lesions, ranging from areas with decreased flow relative to oxygen demand (‘misery perfusion’) to areas with increased flow relative to demand (‘absolute luxury perfusion’) [60]. 

In this context, RA captures subtle variation within medical images and could be used to analyze the heterogeneity of lesions for a better diagnostic or predictive purposes. Regarding the heterogeneity of AIS lesions, radiomics seems to be superior to conventional imaging visual analysis [62]. Texture features allow quantification of the heterogeneity within a lesion by considering both pixel intensity and statistical interrelationship in space (distance or orientation) [12,63,64,65]. Due to their objective and quantitative values, recently, radiomics features were integrated in stroke outcome PMs and compared to clinical based PMs or prognostic scores.

The findings of our systematic review confirmed the superiority of a combined model, suggesting that clinical and imaging factors may have an intercrossing and synergistic effect on each other, resulting in a more satisfactory outcome PM. After combining clinical and radiomics features in their PMs, five [33,35,36,37,39] of the six included studies demonstrated better predictive values compared to models based only on clinical or imaging variables. Therefore, two studies [34,35] performed nomograms, integrating the clinical and radiomics features that have achieved the best results in PMs. Interestingly, the most efficient nomogram [35] resulted after combing more clinical factors such as age, gender, stroke history, diabetes, baseline mRS and NIHSS and less radiomics variables (7 texture features in Zhou et al. study [35] versus 11 texture features in Wang et al. study [34]). This could be explained by the fact that Zhou et al. [35] used multiple feature reduction methods, beginning with 1310 extracted features of different types and applying variable statistics tools (from Spearmen’s correlation to minimum redundancy maximum relevance and least absolute shrinkage and selection operator) to reduce the redundancy of features and to select the best predictive ones. Moreover, among the radiomics features selected, exponential gray level non-uniformity and wavelet feature cluster prominence were the best predictors [35]. Both features quantify the similarity of gray-level intensity values in the image and describe the heterogeneity of the infarcts. Thus, higher values indicate higher signal heterogeneity of the infarcted lesion, the possibility of lesion progression and worse outcomes [66]. These findings are incongruent with data from Boss et al. study [67] which suggested that visually assessed DWI lesion homogeneity could be associated with significantly higher mRS at three months. Thus, quantitative image analysis via radiomics may offer a better description of lesions’ subtle abnormalities or heterogeneity, adding valuable information to conventional imaging markers.

Wang et al. [37] investigated a clinical and radiomics-based model to predict one-year stroke recurrence and obtained an AUC of 0.84 (95% CI, 0.82–0.87) and the mean interval time between the first stroke and stroke recurrence was 167.11 ± 100.08 days. The stroke subtypes were significantly different between recurrence and non-recurrence groups (*p* = 0.003) [37]. Of the 544 large artery atherosclerosis patients, 10.3% of patients repeated the stroke within the first year, and these patients were older than the non-occurrence stroke group (*p* = 0.016). These findings are inconsistent with previous studies [68] that showed a higher risk of recurrent stroke in large artery atherosclerosis despite aggressive medical treatment. In contrast, 11.3% of atrial fibrillation patients had a stroke recurrence within a year, and these patients were younger than those who did not repeat stroke (*p* = 0.036) [37]. The lowest recurrence rate was in the group of patients with small vessel disease, which is consistent with previous data [69].

Incongruent with the other five studies [33,34,35,36,37], the Wang et al. [38] study failed to achieve predictive values for texture features derived from T2-FLAIR and ADC images. This could be explained by the fact that the other studies [33,34,35,36,37] built extensive PMs based on multiple clinical factors, conventional imaging markers and numerous radiomics features, whereas Wang et al. [38] investigated in this study few radiomics features. However, Wang et al. [38] found that ADC-entropy and T2-FLAIR 0.75 quantile were associated with baseline NIHSS (AUC = 0.7, *p* = 0.01). Entropy measures the randomness in the gray level intensities of an image and, visually, an image with higher entropy will appear heterogeneous [70]. 

Assessing the risk of bias using the PROBAST [32] tool, potential high risk of bias in participants selectin in two studies was identified. Wang et al. [37] excluded a large number of patients (577) from the initial sample due to cerebral hemorrhage and previous neurological or psychiatric disorders. Quan et al. [33] also excluded 154 patients due to bilateral cerebral infarction, multiple territories strokes and neurological dysfunction left by previous AIS or other neurological diseases. All of these factors are associated with unfavorable prognosis in AIS patients [71,72], thus, participants selection could influence the findings of the studies. Moreover, only patients with MCA stroke were included in Quan et al. [33] study, thus, their findings cannot be generalized to strokes in other territories. Moreover, the findings of Quan et al. [33] and Wang et al. [37] should be interpreted with caution because the datasets from these studies were imbalanced (number of cases with favorable outcome was much higher compared to number of cases with unfavorable outcome), and oversampling methods were applied such as Synthetic Minority Oversampling Technique (SMOTE) [73] to increase the number of cases in the unfavorable outcome group from both training and validation cohorts. Regarding the participant selection process, it is important to notice that in the population from three studies [33,34,38] prevailed the male participants with at least 60% proportion. Previous research suggested that women are more likely to develop a poor long-term outcome after AIS, having a two-to-three-fold risk of poor outcome compared to men, as women develop AIS at an older age when they have multiple comorbid diseases [74,75]. 

The RQS is a recently introduced score whose aim is to assess the methodological quality of radiomics-based studies [15] and does not consider differences in study objectives. It could help identifying high-quality results among the large number of publications in this field, as well as issues limiting their value and applicability [28]. The median RQS of the studies included in our systematic review is 15, reflecting a moderate methodological quality. This finding is consistent with previous systematic reviews performing quality assessment with RQS tool in other fields of neuroradiology [28,29,30]. However, the RQS score was relatively recently introduced and has been applied in a limited number of occasions [15,76,77,78]. In our review, all studies collected 0 points on the following items: imaging at multiple time points, performing a prospective study to apply the model and cost-effectiveness analysis. Therefore, temporal variability was never tested, also due to the retrospective design of studies. 

Our study has some limitations that should be acknowledged. The number of included studies was low, probably due to strict inclusion criteria and pre-established study objectives to assess the role of radiomics in ischemic stroke outcome prediction. Study heterogeneity was moderate and meta-analysis was not possible, but this is in line with other systematic reviews investigating RA in the field of neuroradiology [28,29,30,79,80]. However, to our knowledge, this is the first systematic review evaluating the role of radiomics in stroke outcome assessment and applying the quality radiomics score in stroke studies.

AI technologies will herald fundamental changes in healthcare delivery [81], providing patients with optimal tailor-made treatment. Radiomics may prove to be one of the most impactful AI applications by bridging the gap between medical imaging and personalized medicine [15]. Radiomics-based tools have the potential to change clinical practice in AIS management by exploring digitally encrypted imaging information related to cerebrovascular pathophysiology. Radiomics integrated in AI algorithms could improve stroke diagnosis in acute phase (e.g., diagnosis of acute lacunar stroke on CT, prediction of hemorrhagic transformation) [46] or in chronic phase (e.g., MRI radiomics features may identify prior or undocumented stroke lesions) [19], guiding the secondary prevention strategies. Machine learning algorithms based on radiomics features also seem to be promising tools for assessing collateral circulation status [49] or clot composition [51], providing important data that could affect the decision for mechanical recanalization techniques. Developing stroke outcome predictive scores based on clinical and quantitative imaging information and improving them in clinical settings, long-term post-stroke disability could be more accurately assessed, helping physicians to create personalized rehabilitation strategies. However, to create tools with clinical utility, prospective trials that validate radiomics signatures on external datasets are required [81]. There is also a need for standardization of RA in line with recent recommendations [16]. Moreover, identification of radiomics features that remain robust, especially against differences in image acquisition and reconstruction from different scanners, needs further research. 

## 5. Conclusions

Our findings suggest that combined models integrating both clinical and advanced imaging variables seem to better predict the patients’ disability outcome group (favorable outcome: mRS ≤ 2 and unfavorable outcome: mRS > 2) at three and six months after stroke onset. Radiomics may be successfully used in AIS assessment, treatment selection and long-term prognosis, providing patients with optimal tailor-made management. In our review, moderate methodological quality of AIS radiomics studies was identified. External validity, prospective studies, cost-effectiveness analysis and publicly available RA protocols are needed to increase methodological quality in stroke radiomics studies. Although their predictive values are significant in the research field, radiomics-based PMs should be validated in multiple clinical settings to become relevant prognosis tools in daily clinical practice and to promote personalized precision medicine.

## Figures and Tables

**Figure 1 diagnostics-13-00857-f001:**
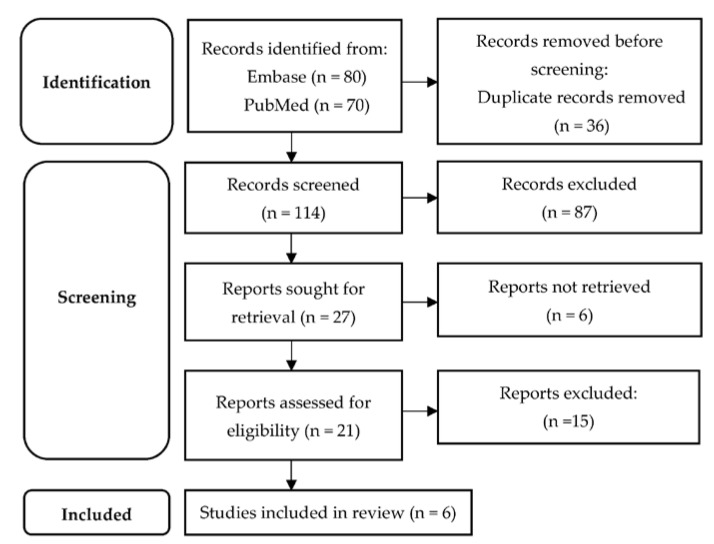
PRISMA diagram of the study selection process.

**Figure 2 diagnostics-13-00857-f002:**
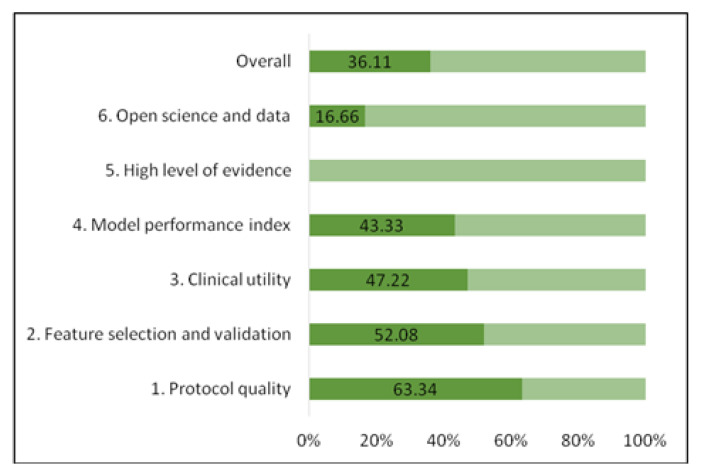
Adherence rate of the RQS of the included studies according to RQS key domains.

**Figure 3 diagnostics-13-00857-f003:**
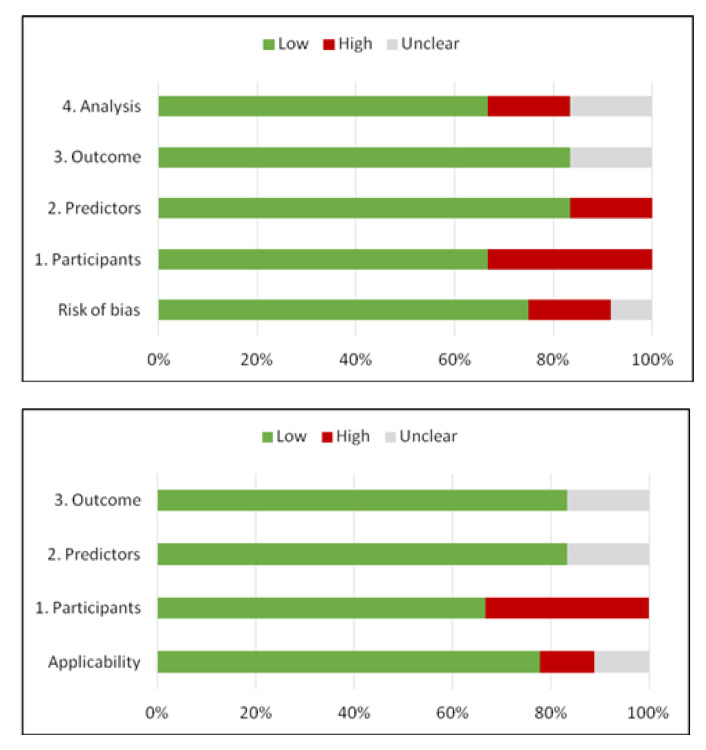
Methodological quality of the studies included according to PROBAST tool for risk of bias and applicability concerns.

**Table 1 diagnostics-13-00857-t001:** Inclusion and exclusion criteria.

Inclusion Criteria	Exclusion Criteria
Studies that investigated MRI radiomics features in patients with AIS Studies that assessed the clinical outcome based on RA features in AIS patients	Unavailable data on RA and predictive model performance CT-, CTA- or US-based RA studies Non-original investigations (reviews, editorials, letters or opinions)

CT = computer tomograph, CTA = CT angiography, US = ultrasound.

**Table 2 diagnostics-13-00857-t002:** Characteristics of studies included in systematic review.

Study, Year	Sample, Age, Sex	AIS Type	Tx	Onset-to-MRI Time	Outcome Criteria	Clinical Factors	MRI Markers	MRI Seq	RA Features	Predictive Models	AUC, 95% CI
Quan et al. [33], 2021	110, 62, 70.9% male	first AIS in MCA territory, onset ≤72 h	ivT, MT: 12 p	26.5 ± 15.7	90 days unfavorable outcome mRS > 2	Age, gender, admission NIHSS	DWI-ASPECT score, ODs	FLAIR ADC	6, TA, wavelet	Clinical	0.79, 0.68–0.89
										Clinical + MRI	0.78, 0.68–0.88
										ADC radiomics	0.77, 0.62–0.83
										FLAIR radiomics	0.73. 0.62–0.83
										ADC + FLAIR radiomics	0.81, 0.73–0.89
										RA + Clinical + MRI	0.92, 0.87–0.97
Wang et al. [34], 2021	399, 67, 63.9% male	NR	NR	within 24 h after AIS onset	90 days outcome mRS > 2	Age, 24-h NIHSS	Hemorrhage	DWI	11, TA	Clinical model	0.77, 0.71–0.84
										Radiomics model	0.70, 0.64–0.77
										Clinical + radiomics	0.80, 0.75–0.86
Zhou et al. [35], 2022	311, 58, 72.7% male	Pen artery: 43.1%, cMCA: 28.6%, cACA: 5.5%, cPCA = 8.4%, ≥2 territories: 14.5%	NR	<24 h: 6.1%24–72 h: 93.9%	6-month good outcome (mRS ≤ 2), poor outcome (mRS > 2)	Age, gender, stroke history, DM, b-mRS, b-NIHSS	-	DWI, ADC	7, first-order statistics, TA	Clinical model	0.82, 0.77–0.87
										Radiomics model	0.76, 0.70–0.82
										Clinical + radiomics	0.86, 0.82–0.91
Zhang et al. [36], 2022	103, 65, 64% male	Unilateral anterior circulation	NR	NR	90 days outcome mRS > 2	Atrial fibrillation	-	ADC	7, TA, wavelet, LGT	ADC	0.60, 049–0.71
										tADC	0.83, 075–0.91
										tADC + clinical	0.86, 079–0.93
Wang et al. [37], 2022	1003, 67, 67.9% m	Ant-circ: 68.5%, Post-circ: 28.5%, Both: 3%	NR	72 h of AIS onset	90 d outcome 1y AIS recurrence	NR	-	DWI	100, TA, wavelet	Radiomics model	0.77, 0.75–0.80
										Clinical + radiomics	0.84, 0.82–0.87
Wang et al. [38], 2020	116, 64, 72% male	NR	NR	NR	90 days outcome mRS > 2, stroke severity	-	-	FLAIR, ADC	15, first-order statistics, TA	RA features were not predictive of mRS. ADC-entropy and T2-FLAIR 0.75 quantile predicted AIS severity (AUC = 0.7, *p* = 0.01).	

Tx = treatment, MRI Seq = MRI sequences for feature selection, MCA = middle cerebral artery, ivT = intravenous thrombolysis, MT = mechanical thrombectomy, OD = orthogonal diameters, TA = texture analysis, FLAIR = fluid-attenuated-inversion recovery, ADC = apparent diffusion coefficient, TA = texture analysis, tADC = texture analysis from ADC, Pen artery = penetrating artery, cor-MCA = cortical branches of middle cerebral artery, cor-ACA = cortical branches of anterior cerebral artery, cor-PCA = cortical branches of posterior cerebral artery, DM = diabetes mellitus, b-mRS = baseline mRS, b-NIHSS = baseline NIHSS, LGT = Laplacian of Gaussian transformation, NR = not reported.

## Data Availability

Data is contained within the article or Appendix A. The data presented in this study are available in [insert article or Appendix A here].

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
