# Peer review of "MRI Radiomics and Predictive Models in Assessing Ischemic Stroke Outcome—A Systematic Review"

_diagnostics, 2023, doi:10.3390/diagnostics13050857_

Round 1

Reviewer 1 Report

Very interesting article about the importance of radiomics analysis (RA). The authors conducted a meta-analysis of the assessment of the methodological quality of radiomics studies, and the radiomics quality index (RQS) was also used. Among many papers, the authors have thoroughly analyzed the importance of prognostic scales and systems for assessing the severity of brain tissue damage in patients with acute stroke, which aim to estimate the likelihood of a good treatment outcome.

The analysis covered the work until the end of 2022, so it is very up-to-date, without a reference in the existing literature.

The authors finally emphasize that the integration of clinical and imaging variables improves the reliability of predictive models, which may translate into better estimation of the predicted course and outcome of stroke.

The downside is the use of MRI-only data. In everyday practice, imaging tests in patients with a stroke are performed with the use of a CT scan and the analysis of the results obtained in such a study.

Perhaps, however, data showing the significance, also prognostic, of MRI examinations will be a contribution to the development of new algorithms for MRI examinations in patients with stroke.

Author Response

Thank you very much for your assessment.

Reviewer 2 Report

Interesting and well written study regarding the role of radiomionics in predicting the outcome of patients with acute ischemic stroke.

The study deserves interest since the the possibilities of radiomics are promising, as are its future clinical applications.

I would suggest in the bract to be more specific regarding the type of research you led, and above all which are the main conclusions we can draw from it, especially as regards the routine clinical application.

Just some minor comments regarding discussion and conclusion: 

- I would suggest to add some lines regarding artificial intelligence tools in the diagnosis of acute ischemic stroke. There are, in fact, nowadays a lot of interesting softwares which are routinely used in common clinical practice to find the site of occlusion, to evaluate the possible collateral circulation, and to analyze perfusion brain patterns, in order to distinguish ischemic core to penumbra. In this context you could comment on that (look for example at the color-via tool: Verdolotti T. et al. ColorViz, a New and Rapid Tool for Assessing Collateral Circulation during Stroke. Brain Sci. 2020 Nov 20;10(11):882. doi: 10.3390/brainsci10110882. PMID: 33233665; PMCID: PMC7699692. ). I think this could add value to your study and to your findings. 

- Do you think radiomics will change clinical practice in acute ischemic stroke management? can you comment on that?

- Conclusion: can you better specify which are the main findings of your study?

Author Response

REVIEWER NO. 2

[Comment 1] I would suggest in the bract to be more specific regarding the type of research you led, and above all which are the main conclusions we can draw from it, especially as regards the routine clinical application.

[Response] Thank you for highlighting these aspects. We formulated the abstract more accurately and we added in the Materials and Methods section (row 113) the link where the review protocol has been registered (https://osf.io/9dx6j/).

Abstract: Stroke is a leading cause of disability and mortality, resulting in substantial socio-economic burden for healthcare systems. With advances in artificial intelligence, visual image in-formation can be processed into numerous quantitative features in an objective, repeatable and high-throughput fashion, in a process known as radiomics analysis (RA). Recently, investigators have attempted to apply RA to stroke neuroimaging in the hope of promoting personalized precision medicine. This review aimed to evaluate the role of RA as adjuvant tool in the prognosis of disability after stroke. We conducted a systematic review following the PRISMA guidelines, searching PubMed and Embase using the keywords: ‘magnetic resonance imaging (MRI)’, ‘radiomics’, and ‘stroke’. The PROBAST tool was used to assess the risk of bias. Also, radiomics quality score (RQS) was applied to evaluate the methodological quality of radiomics studies. From the 150 abstracts returned by electronic literature research, 6 studies fulfilled the inclusion criteria. Five studies evaluated predictive value for different predictive models (PMs). In all studies, the combined PMs consisting of clinical and radiomics features have achieved the best predictive performance compared to PMs based only on clinical or radiomics features, the results varying from an area under the ROC curve (AUC) of 0.80 (95% CI, 0.75-0.86) to an AUC of 0.92 (95% CI, 0.87-0.97). The median RQS of the included studies was 15, reflecting a moderate methodological quality. Assessing the risk of bias using PROBAST, potential high risk of bias in participants selection was identified. Our findings suggest that combined models integrating both clinical and advanced imaging variables seem to better predict the patients’ disability outcome group (favorable outcome: modified Rankin scale (mRS) ≤ 2 and unfavorable outcome: mRS > 2) at three and six months after stroke. Although, radiomics studies’ findings are significant in research field, these results should be validated in multiple clinical settings in order to help clinicians to provide individual patients with optimal tailor-made treatment.

Also, we added some lines in the Conclusion section (rows 460-476) in order to expose our main findings more precisely (Please see the response to comment 4).

[Comment 2] I would suggest adding some lines regarding artificial intelligence tools in the diagnosis of acute ischemic stroke. There are, in fact, nowadays a lot of interesting softwares which are routinely used in common clinical practice to find the site of occlusion, to evaluate the possible collateral circulation, and to analyze perfusion brain patterns, in order to distinguish ischemic core to penumbra. In this context you could comment on that (look for example at the color-via tool: Verdolotti T. et al. ColorViz, a New and Rapid Tool for Assessing Collateral Circulation during Stroke. Brain Sci. 2020 Nov 20;10(11):882. doi: 10.3390/brainsci10110882. PMID: 33233665; PMCID: PMC7699692. ). I think this could add value to your study and to your findings. 

[Response] Thank you very much for bringing into attention the multiple applications of artificial intelligence tools in acute ischemic stroke diagnosis and management pathway. We added a paragraph regarding the types of AI technologies and its additional roles in stroke diagnosis and assessment (rows 303-331).

The current applications of AI in AIS field seem to be efficient in numerous parts of the diagnostic and management pathways, including detection, triage, and outcome prediction [45]. Computer-aided detection scheme based on texture features from areas known to show early AIS signs such as insula ribbon and lentiform nucleus were suggested to be a promising algorithm for lacunar AIS diagnosis [46]. As lacunar AIS is relatively difficult to diagnose on CT within the first hours after onset [47], early detection is crucial for establishing the best treatment and there is a need for a more efficient method to improve CT detection rate. Automated color maps (e.g., ColorViz) have been proved to be rapid and accurate post-processing tools that permit maintenance of the temporal resolution of CT angiography, summing in a single image the three different cerebral vascular phases using a time variant color map [48,49]. As the definition of the collateral circulation status is essential in selecting patients for mechanical thrombectomy, the possibility of using an immediate scoring scale for CT angiography could make the diagnostic assessment faster and easier. A recent systematic review [50] showed that AI-based comprehensive platforms (e.g., Brainomix, iSchemaView, Viz.ai) could automatically detect the presence of large vessel occlusion (LVO), being a catalyst for timely LVO detection and an aid to acute management decision-making. Moreover, automated clot composition analysis systems using machine learning seem to provide information on the cause of cerebral artery occlusion and may further guide acute revascularization and secondary prevention. For example, a recent study [51] assessed the accuracy of a such algorithm based on blooming effect on pre-treatment gradient echo images (GRE) from 67 patients with middle cerebral artery stroke and identified atrial fibrillation with high accuracy (AUC>0.87). Blooming artifact caused by paramagnetic materials in GRE images has been associated with cardioembolic stroke [52,53], cardioembolic clots having significant higher proportion of red blood cells compared with noncardiac clots and, oxyhemoglobin in erythrocytes goes through sequential stages of degradation into deoxyhemoglobin and hemosiderin, which are paramagnetic materials [54,55].

[Comment 3] Do you think radiomics will change clinical practice in acute ischemic stroke management? can you comment on that?

[Response] Thank you for highlighting this aspect. We added some lines regarding the potential translation of radiomics studies to clinical practice and advances that should be made to overcome some limitations of radiomics analysis (rows 438-457).

AI technologies will herald fundamental changes in healthcare delivery [81], providing patients with optimal tailor-made treatment. Radiomics may prove to be one of the most impactful AI applications by bridging the gap between medical imaging and personalized medicine [15]. Radiomics-based tools have the potential to change clinical practice in AIS management by exploring digitally encrypted imaging information related to cerebrovascular pathophysiology. Radiomics integrated in AI algorithms could improve stroke diagnosis in acute phase (e.g., diagnosis of acute lacunar stroke on CT, prediction of hemorrhagic transformation) [46] or in chronic phase (e.g., MRI radiomics features may identify prior or undocumented stroke lesions) [19], guiding the secondary prevention strategies. Also, machine learning algorithms based on radiomics features seem to be promising tools for assessing collateral status [49] or clot composition [51], providing important data that could affect the decision for mechanical recanalization techniques. Developing stroke outcome predictive scores based on clinical and quantitative imaging information and improving them in clinical settings, long-term post-stroke disability could be more accurately assessed, helping the physicians to create personalized rehabilitation strategies. But to create tools with clinical utility, prospective trials that validate radiomics signatures on external datasets are required [81]. Also, there is a need for standardization of RA in line with recent recommendations [16]. Moreover, identification of radiomics features that remain robust, especially against differences in image acquisition and reconstruction from different scanners needs further research.

[Comment 4]  Conclusion: can you better specify which are the main findings of your study?

[Response] Thank you for bringing into attention this aspect. We formulated more accurately, and we added some lines in Conclusion section to expose all our main conclusions (rows 459-476).

Our findings suggest that combined models integrating both clinical and advance imaging variables seem to better predict the patients’ disability outcome group (favorable outcome: mRS ≤ 2 and unfavorable out-come: mRS > 2) at three months, respectively at six months after stroke onset. Radiomics may be successfully used  in AIS assessment, treatment selection and long-term prognosis, providing patients with optimal tailor-made management. In our review, moderate methodological quality of AIS radiomics studies was identified. External validity, prospective studies, cost-effectiveness analysis and publicly available RA protocols are needed to increase methodological quality in stroke radiomics studies. Although, their predictive values are significant in research field, radiomics-based PMs should be validated in multiple clinical settings to become more relevant prognosis tools in daily clinical practice and to promote personalized precision medicine.

The references were automatically modified (Zotero) as new citations were included.
